# Insect Microbial Symbionts: Ecology, Interactions, and Biological Significance

**DOI:** 10.3390/microorganisms11112665

**Published:** 2023-10-30

**Authors:** Sankhadeep Mondal, Jigyasa Somani, Somnath Roy, Azariah Babu, Abhay K. Pandey

**Affiliations:** 1Deparment of Entomology, Tea Research Association, Tocklai Tea Research Institute, Jorhat 785008, Assam, India; sankhadeep810@gmail.com (S.M.);; 2Deparment of Mycology & Microbiology, Tea Research Association, North Bengal Regional R & D Centre, Nagrakata, Jalpaiguri 735225, West Bengal, India

**Keywords:** gut symbiont, bacteria, gut communities, immune system, symbiosis, mutualism

## Abstract

The guts of insect pests are typical habitats for microbial colonization and the presence of bacterial species inside the gut confers several potential advantages to the insects. These gut bacteria are located symbiotically inside the digestive tracts of insects and help in food digestion, phytotoxin breakdown, and pesticide detoxification. Different shapes and chemical assets of insect gastrointestinal tracts have a significant impact on the structure and makeup of the microbial population. The number of microbial communities inside the gastrointestinal system differs owing to the varying shape and chemical composition of digestive tracts. Due to their short generation times and rapid evolutionary rates, insect gut bacteria can develop numerous metabolic pathways and can adapt to diverse ecological niches. In addition, despite hindering insecticide management programs, they still have several biotechnological uses, including industrial, clinical, and environmental uses. This review discusses the prevalent bacterial species associated with insect guts, their mode of symbiotic interaction, their role in insecticide resistance, and various other biological significance, along with knowledge gaps and future perspectives. The practical consequences of the gut microbiome and its interaction with the insect host may lead to encountering the mechanisms behind the evolution of pesticide resistance in insects.

## 1. Introduction

Insects are the most diverse category of creatures, with over a million species inhabiting almost every environment [1]. Owing to their worldwide spread, insects relate to a variety of microorganisms, including bacteria, viruses, fungi, protozoa, nematodes, and multicellular parasites [1]. Under certain circumstances these microbes make diverse interactions with eukaryotic organisms, including insects [2]. These relationships might be symbiotic, pathogenic, or vectorial. Insects have a strong relationship with their gut microbiota and this symbiotic relationship has become a critical evolutionary adaptation for their survival in harsh environments [2]. The microbiota of the insect stomach exemplifies all microbial associations, from harmful to obligatory mutualism. The emphasis on research of insect–bacteria symbiosis has been superseded by a focus on insect–microbial pathogen connections and the development of microbial pesticides [2]. The relevance of bacteria existing inside insects has been explored in the context of advancing our knowledge of symbiotic connections and multitrophic interactions between arthropods and gut bacteria; this will aid in the development of novel tactics for insect pest management [2,3]. The major focus of insect microbiology is the interactions between insects and harmful (as well as beneficial) microbes. In insects, gut symbiotic microorganisms play a variety of physiological roles in host metabolism, including digestion of food, providing essential nutrients that are scarce in food, preventing pathogen invasion by stimulating the host immune system, degradation of phytotoxins and pesticides, production of antibiotics, and competition for limited nutrients [3]. Some experts believe these symbionts to be “intracellular parasites” that have seized control of the insect’s body and developed numerous methods to secure their survival while delivering advantages to their host. Yet, it is also possible that the insect created this interaction with its microbiota to ensure its own survival. Whatever the case, they have now accommodated themselves to each other [4].

Bacterial species inside an insect’s digestive tract might represent mutualism, commensalism, or parasitism [4]. Microorganisms are engaged in several host life, physiological, and evolutionary processes, including feeding, reproduction, immunological homeostasis, defense, and speciation. Hence, the modification and elevated utilization of microbiota is an essential application tool for the creation of methods for the mitigation of insect-related issues [3,4]. This method, termed “Microbial Resource Management” (MRM), has been effectively implemented in several habitats and ecosystems [5]. Even though prior researchers have published studies on insect gut endosymbionts as biotechnological resources in agriculture [5,6,7], they have not highlighted their multifaceted roles such as mode of symbiosis, their role in insecticide resistance, or biological significance. This present review bridges the gap between insect symbiosis research and insect pest management by providing a complete overview of the scenarios in which insect symbiosis research may help to manage the insect pests. Insect ecology and symbiosis knowledge help us to target systems with intriguing symbionts that are prospective sources of insect pest control [6]. In addition, we analyze the inner-gut microbial populations and their significance in pesticide resistance management, as well as information gaps and possible future perspectives. Furthermore, this review also highlights how insects use their gut symbiont to live in a variety of harsh environments and how gut symbionts can be used to reduce insecticide resistance problems.

## 2. Microbial Symbionts in Insects

In insect–microbe interactions, the bacterial component is the symbiont and the insect part is the host. Symbionts are typically microbes such as bacteria, archaea, and fungi [5]. Symbionts consist mostly of primary and secondary symbionts that interact intimately with the host insect [8]. Primary symbionts provide a crucial function for the insect and are maternally transferred from one generation to the next [8] **(**Figure 1). In insects such as aphids, tsetse flies, and psyllids, the primary symbiont performs a protective role and synthesizes vital nutrients [5]. Whiteflies, *Bemisia tabaci* Gennadius (Hemiptera:Aleyrodidae), for instance, host *Candidatus Portieraaley rodidarum* Costa (hereafter *Portiera* sp.) as their major symbiont. Like the *Portiera* sp., some other major symbionts of phloem-feeding insects provide essential carotenoids and amino acids for their whitefly hosts [9]. Another example of a major symbiont is *Buchnera aphidicola* Munson et al. (Enterobacterales: Erwiniaceae), which is found inside bacteriocytes in the abdominal body cavity of almost all aphids and contributes necessary amino acids that are absent from the insects’ phloem sap diet [10]. In contrast, secondary symbionts are typically diverse and exhibit less host-specific commitments [11]. Some bacteria, known as parasites or pathogens, are detrimental to their host, while others, known as mutualists, are beneficial to their host. Mutualistic interactions are common in insects; they create a wide diversity of alliances with microorganisms. Symbiotic bacteria have a crucial role in high-temperature tolerance, resistance to parasitoids, protection against harmful viruses, and toxin synthesis [11]. These microbes play a significant role in the development of insects by enhancing their adaptability to heterogeneous environments [8]. For instance, the *Rickettsia* sp. improves basic host fitness [12] and helps the host to survive against *Pseudomonas syringae* Van Hall (Pseudomonadales: Pseudomonadaceae), an entomopathogen in whiteflies [13]. The bacterium *Hamiltonella defensa* Moran et al., alters the host sex ratio by supplying whitefly with nutrients [14].

## 3. Types of Insects–Bacterial Interaction

The association between insects and microorganisms may be symbiotic or harmful. A symbiotic relationship involves the strong connection of two distinct species [15]. This type of relationship is classified as commensalism, mutualism, or parasitism based on the fitness impacts of the interaction on its members [15]. The clearest example of commensalism is exhibited in the bark beetle galleries, which offer nutrients and shelter for a range of insects and microbial commensals, most of which have slight or no impact on the bark beetles [16]. When one species gains fitness at the cost of the others, it is said to be parasitic [4]. In mutualism, for example, numerous situations of essential association are needed for the survival and reproduction of few insects, such as termites and their protozoan and bacterial companions [17]. Pathogenic interactions are described as the means through which microorganisms survive and interact inside their host organisms [17]. In this relationship, the pathogen must be highly specialized and have a close touch with the host. For instance, fungi are the most prevalent disease-causing agents of insects and are also essential to the health of natural ecosystems [18].

In particular, these symbionts are either obligate primary (P) endosymbiont, with a long evolutionary relationship with their hosts and are essential for their fertility and survival, or facultative secondary (S) symbiont, which have formed a more recent association with the host and have retained the ability to return to a free-living state [6]. P-endosymbionts are maternal vertical transmissions to the offspring (in the egg) and enclosed in specialized cells known as bacteriocytes (or mycetocytes), which are sometimes aggregated into organ-like structures called bacteriomes (or mycetomes). An endosymbiont identified in the *Sitophilus oryzae* L. (Coleoptera: Curculionidae) (rice weevil) (2 × 10^3^ bacteria per host cell) is known as the SOPE or *S. oryzae* primary endosymbiont [19]. The cytoplasm of the bacteriocyte contains an Enterobacteriaceae family bacterium, whose expression is somewhat controlled by the host. This bacterium serves as the insect’s source of a number of vitamins, including biotin, pantothenic acid, and riboflavin [19]. S-symbionts, on the other hand, are often transported vertically despite horizontal transmission [15]. S-symbionts have various effects on their hosts. For instance, *Rickettsia* sp. improves host fitness generally and increases host resistance to *P. syringae* (an entomopathogen found in whitefly) [12,13]. *Hamiltonella defensa* (Enterobacterales: Enterobacteriaceae), a rare endosymbiont of sap-sucking insects that protect hosts from parasitic wasp attacks, is another example of S-symbiont [20].

Insects contain a broad variety of bacteria in their digestive tracts. This microbe gives physiological and ecological benefits to its host. Gut symbionts of the genus *Citrobacter* (Enterobacterales: Enterobacteriaceae) that were isolated from the gut of *Lepidiota mansueta* Burmeister (Coleoptera: Scarabaeidae), the root-feeding white grub beetle [21], participate in the detoxification of phytophagous insects and aid in the degradation of lignocellulose in xylophagous insects, as well as provide protection from pathogens by producing an antimicrobial substance [22]. Many different interactions exist between insects and bacteria. Insects rely on bacterial symbionts for several crucial functions. Symbiotic bacteria are crucial for the lifestyle of the host and development [23] through food digestion, the production of energy and vitamins, and the formation of the body’s natural defenses [24]. It has been demonstrated that microbial symbionts have several effects on insect behavior and health [25]. Some insects have specialized organs for retaining a small number of symbiont species, while others contain varied and fluctuating flora in their digestive tracts and other internal organs. Numerous connections are formed with one or a few bacterial species (Table 1). These may include the establishment of specialized insect structures and cells, such as midgut crypts, mycangia, and microbiomes, in order to keep certain obligatory symbionts [26]. For instance, several phytophagous stinkbugs have obligatory symbionts in a specialized midgut area that comprises multiple crypts [26]. Recent research revealed that the midgut crypts of the *Largidae* family of the superfamily *Pyrrhocoroidea* harbor a *Burkholderia cepacia* Yabuuchi et al. (Burkholderiales: Burkholderiaceae) symbiont [27]. In fact, midgut stem cell mediated response to bacteria-induced tissue damage has been studied in *Drosophila melanogaster* Meigen [28,29,30], along with crucial function of proliferation of midgut cells in the establishment of insect vector competence [31]. Several antibiotic-producing actinomycete symbionts have been identified from pine beetles and their mycangium [32].

## 4. Habitat of Microorganisms within Insect Gut

In insects, the exoskeleton that lines the foregut and hindgut, which are both generated from embryonic ectoderm, is made up of cuticular glycoproteins and chitin [46]. This exoskeleton separates the intestinal lumen from the epidermal cells and is shed at each ecdysis [26]. Insect foreguts frequently have distinct crops or diverticula for short-term food storage and their hindguts have distinct sections, e.g., fermentation chambers and recta that are used to store feces prior to defecation [26]. For instance, *Cyclocephalla signaticollis* Burmeister (Coleoptera: Scarabaeidae) has a typical alimentary tract with a modified extended part of the hindgut known as the fermentation chamber [47]. The digestive system of scarab larvae is populated by a variety of microbes, the majority of which are concentrated in a fermentation chamber and are crucial for the digestion of plant matter [47].

Many insects’ primary location of digestion and absorption is their midgut, which is made of endodermal cells and lacks a cuticle. The midgut epithelial cells of numerous insects develop a covering known as the peritrophic matrix (or peritrophic membrane or PM). The midgut is divided into the endoperitrophic space and the ectoperitrophic space by the peritrophic matrix. In order to avoid direct contact with the midgut epithelium, the microbes are often limited to the former [48]. The two basic types of peritrophic matrix are type I peritrophic matrix and type II peritrophic matrix [48]. In many lepidopterans, several coleopterans, certain dictyopterans, and orthopterans, as well as some hymenopterans, are examples of insects that have a type I peritrophic matrix. The type II PM is present in a few lepidopteran and certain dipteran insects [49]. A number of different functions are carried out by the peritrophic matrix. In addition to concentrating food and digesting enzymes, it also acts as a barrier to prevent the epithelium from being exposed to large toxin molecules found in food, from food particles inflicting mechanical injury, and from microbial invasion [50]. The majority of symbiotic relationships that detoxify the body are referred to as “gut symbioses,” in which symbiotic microorganisms are extracellularly contained in the lumen of the digestive tract. The stinkbug species *Murgantia cribraria* Fab. and the bean bug species *Riptortus pedestris* Fab. both generate a number of sac-like structures towards the posterior part of the midgut. These symbiotic bacteria are contained in this sac [8]. In the plataspid stinkbug, the bacteria localize in midgut crypts [51].

## 5. Mechanism of Transmission of Gut Symbionts

In host insects, symbionts are vertically passed from one generation to the subsequent. The host can maintain a symbiosis across several generations due to symbiont transmission [52]. During the transmission of gut symbionts, two primary mechanisms are observed. Horizontal transmission occurs when the symbionts are transferred from one individual to another, typically by contact with bodily fluids or excretions [52]. Vertical transmission occurs when the host’s progeny can acquire the symbionts from the parents (transmission of symbionts from mother to offspring) (Figure 2). There are several variants of these two modes and transmission may also entail vertical and horizontal transfers, as well as intraspecific or interspecific host switching [52]. *Serratia symbiotica* Moran et al. (Enterobacteriales: Yersiniaceae), a bacterial species that lives as a mutualistic symbiont of aphids, is vertically transferred inside the mother’s body by transovarial endocytosis [52]. Researchers have discovered that a culturable strain of *S. symbiotica* has a greater collection of ancestral *Serratia* genes, is a gut pathogen in aphid hosts, and is mostly transmitted through a fecal–oral pathway [52]. The majority of directly documented horizontal symbiont transmission events involve parasitoids, as is the case when *Arsenophonus nasoniae* Kirkaldy et al., uninfected parasitoids receive the infection while maturing in the same host as their infected counterparts [53].

Symbionts travel a vast distance from the point of first contact to their eventual destination. After contacting the host, the organism enters the body of the host, slips past the immune system, and then makes its way to the organ that houses the symbiont [54]. Fruit and olive flies harbor symbiotic bacteria in their guts and pharyngeal bulbs; they spread the symbionts via contaminated egg surfaces. The kudzu bug, *Megacopta cribraria* Fab. (Hemiptera: Plataspidae), has a complex and unusual method of symbiont transfer known as “capsule transmission”. Mother bugs deposit symbiont-filled particles called “symbiont capsules” with eggs and hatchlings explore the capsules to receive the symbiont, through which vertical transmission takes place [8]. However, failure of this vertical transfer may result in the symbiont’s demise unless horizontal transmission happens. Hence, the failure of symbiont transmission may result in lower fitness, slowed development and growth, sterility, or even the death of the host [55]. For instance, stinkbugs (*Plataspidae*) are known for their distinctive vertical transmission mechanism known as the “symbiont capsule”, which houses a bacterial symbiont in the posterior midgut. The host insects exhibit slowed nymphal development when the symbiont is eliminated [56]. Females of the *P. japonicus* Cuvier et al., produced symbiont capsules upon oviposition and the contents of their guts exhibited specific characteristics for capsule formation.

According to phylogenetic research, the obligatory endocellular symbionts of aphids, *B. aphidicola*, are found to form a sister group to the plataspid symbionts in the Proteobacteria [57]. When the symbionts are removed, the insects develop slowly, die, or become sterile. Despite the extracellular connection, the host phylogeny completely coincides with the symbiont phylogeny, demonstrating rigorous host–symbiont co-speciation [57]. *Nezara viridula* (Heteroptera: Pentatomidae), a heteropteran insect that feeds on plants, has also been discovered to harbor a particular symbiont in gut crypts and acquires the symbiont environmentally each generation [58], indicating that environmental transmission is also compatible with the high specificity of a symbiotic relationship. The *Glossina austeni* Newstead (Diptera: Glossinidae) exhibits *Sodalis glossinidius* Aksoy et al., both in various tissues intracellularly and in the gut lumen; transmission is performed as a result of a specific reproductive process in which larvae develop within the maternal uterus and consume milk secretions carrying *S. glossinidius* Aksoy et al. [59].

Social insects, such as termites and social bees, have additional specialized gut symbionts that are transferred vertically [60]. In the first few days after emerging from the pupal stage, adult honeybees, *Apis mellifera* L. (Hymenoptera: Apidae), acquire bacterial symbionts localized to the hindguts via social contacts with other adult worker bees in the colony [60]. The gut communities are more complicated and transmission seems to occur through coprophagy or proctodeal trophallaxis predominantly inside colonies [61]. By defecating and feeding in a shared space, gregarious insects such as cockroaches and crickets may spread bacteria. *Thermobia domestica* Packard et al. (Zygentoma: Lepismatidae) group together in response to certain bacterial species found in their conspecifics’ excrement. As a consequence, these bacteria have been shown to be horizontally transmitted between insects [62,63].

## 6. Composition of Microbiome in Insect Gut

Many bacterial phyla, such as Betaproteobacteria, Bacteroidetes, Firmicutes, including *Lactobacillus* and *Bacillus* species, Gammaproteobacteria, Alphaproteobacteria, Clostridia, Actinomycetes, Spirochetes, Verrucomicrobia, and others, are often found in the guts of insects [64]. The makeup of the gut microbiota is influenced by a variety of factors, including insect development, biochemical changes in various intestinal locations, and the insect’s capacity to access available nutrients [5]. Insect hindguts serve as an extension of their body cavities and are essential for collecting nutritional waste. As a result, they provide the gut microbiota with a perfect feeding environment, encouraging their variety and proliferation [48]. Due to their positioning, intestinal epithelial cells sometimes come into direct contact with the gut microbiota. The endoperitrophic space, which lines the center of the gut, is where most gut bacteria are contained. There are bacteria, fungi, archaea, and protozoa in the microbiota of insect guts [48]. For instance, both the higher and lower termites include bacteria and archaea in their digestive tracts [65]. Researchers have shown that a variety of bacterial species, including *Snodgrassella alvi* Engel et al. (Neisseriales: Neisseriaceae), *Gilliamella apicola* Kwong et al. (Orbales: Orbaceae), *Lactobacillus*, and *Bifidobacterium bifidum* Orla-Jensen et al. (Bifidobacteriales: Bifidobacteriaceae), predominate in the digestive tracts of mature honeybee workers [23].

The microbiome of the insect stomach includes protists, which comprise almost 90% of the hindgut of subterranean termites [26]. The insect gut contains a large amount of food waste, which provides gut microbiota with a nutrient rich environment [66]. Protists are most commonly found in lower termites and wood roaches; they depend on social transmission [65]. Fungi are mostly found in the insect gut; they feed on wood or detritus. Insects that consume wood or debris, such as beetles and termites, keep methanogenic archaea in their guts [67]. A variety of bacterial species are present in the guts of many insect species. Most of the studies, however, depend on bacterial 16S rRNA gene primers, which can influence perceptions of the makeup of insect gut ecosystems. The bacterial communities in the gut differ greatly in terms of their overall size, composition, distribution, and functions [67]. For instance, an adult grasshopper (*Melanoplus sanguinipes* Fab.) has roughly 10^6^ bacteria compared with the 10^9^ bacteria found in adult *Rhodnius prolixus* Stal et al. (Hemiptera: Reduviidae) [68]. In adult honeybees, 10^9^ bacterial cells are also seen [60]. An adult fruit fly (*Drosophila melanogaster* Meigen) contains approximately 10^5^ microorganisms [69,70]. The changes in insect species may be the cause of the variations in the bacterial community in the gut. Other significant differences in gut communities include microbes that are especially suited to live in insects or that may be passed directly between hosts or that are acquired each generation from the outside environment [69]; these transmission characteristics are often connected. Bacteria obtained from the environment may be harmful or symbiotic [71,72].

## 7. Detection and Diagnosis of Gut Bacteria

Insect guts are home to extensive and diverse microbial ecosystems. There are many conventional methods for identifying gut bacteria, including the use of selective media such as Peotone yeast extract agar, biochemical assays, and species-specific kits. In a previous work, Lloyd et al. [73] used culture, morphological, and biochemical techniques to identify bacterial species in the intestine of *Bactrocera zonata* Saunders et al. (Diptera: Tephritidae), a peach fruit fly. Using similar methods, Naaz and Choudhary [74] discovered three prominent bacterial symbionts, including *Rhodococcus* spp., *Klebsiella oxytoca* Flugge, and *Microbacterium* spp. In addition, Luria–Bertani agar medium was used to identify specific gut symbionts, as well as Gram staining and biochemical methods [75]. Nevertheless, these methods are unreliable to some degree. Thus, microbiologists have been relying on molecular methods to identify different bacterial species in recent years [76]. With the use of 16S RNA sequencing, Deli et al. identified *Bacillus* sp., *Paenibacillus* sp. (Bacillales: Bacillaceae)*, Acinetobacter lwoffii* Bouvet et al. (Pseudomonadales: Moraxellaceae), *Staphylococcus* (Bacillales: Staphylococcaceae), and *Exiguobacterium acetylicum* Collins et al. (Bacillales: Bacillaceae) associated with guts of springtails. Walker et al. [77] used morphological, biochemical, and 16S rRNA studies to identify gut bacteria, including *P. putida* Trevisan et al., *Delftia acidovorans* Vaneechoutte et al. (Burkholderiales: Comamonadaceae), *Defluvibacter*, *Flavobacterium johnsoniae* Bergey et al. (Flavobacteriales: Flavobacteriaceae), and *Ochrobactrum anthropi* Holmes (Hyphomicrobiales: Brucellaceae) from the fruit fly *Bactrocera tau* Walker (Diptera: Tephritidae).

## 8. Influence of Gut Bacteria on the Activity of Pesticides

Insects harbor a variety of bacteria in their digestive tracts, which provide physiological and ecological benefits. The microbial communities associated with the insect digestive tract are very dynamic and involve the elimination of several stresses. The microbiota in insect guts is subject to strong selection pressure and is impacted by factors including food shortages, dietary changes, and exposure to toxins [78]. Insect microbiota exposed to pesticides may also help the hosts to digest these hazardous chemicals (Figure 3). Moreover, it also serves as a source of diversity, which lowers the host’s vulnerability to pesticides [79]. The digestive system of an insect includes a variety of microbial communities belonging to the phyla Firmicutes, Proteobacteria, and Actinobacteria that have a significant effect on the biology of the host [80].

Many studies have revealed that bacteria in the intestines of insects break down several insecticides and reduce their efficiency [76]. Moreover, cultivable gut bacteria have a wide range of consequences for pest control tactics. For example, bacteria are shown to be involved in the breakdown of harmful components eaten by the host insect, leading to pesticide resistance [81,82]. The specific components of these bacterial odors have a significant impact on fruit fly behavior as either feeding or ovipositional stimulants [83] and are utilized in pest control as traps or baits [84]. The Diamondback moth, *Plutella xylostella* L. (Lepidoptera: Plutellidae), a widespread pest of the brassica crop, has been discovered to be resistant to all classes of pesticides in different research studies. This insect is home to a variety of microbiota that aid in the enzymatic breakdown of xenobiotics such as pesticides [76]. The insect’s stomach *Proteobacteria* can degrade acephate, lambda-cyhalothrin, trichlorfon, chlorpyrifos, and spinosad [85].

For instance, *Burkholderia cepacia* symbionts have been shown to increase insecticide resistance in *Riptortus pedestris* Fab. and fenitrothion-degrading *Burkholderia* strains may also be horizontally transferred to other insects [86]. The same study discovered that the trichlorfon-degrading *Citrobacter freundii* Werkman and Gillen (CF-BD) isolated from the stomach of *B. dorsalis* boosts pesticide resistance in the cockroach gut [24]. Hence, the various gut bacteria found in various insects perform a crucial role in insecticide degradation and help to establish resistance in the physiology of insects. Three proteobacterial families, including *Enterobacteria*, *Pseudomonas*, and *Burkholderia*, facilitate the breakdown of these insecticides [24,80]. The resistant strain of *Spodoptera frugiperda* Smith (Lepidoptera: Noctuidae), which is able to break down insecticides, primarily spinosad, chlorpyrifos, deltamethrin, lambda-cyhalothrin, and lufenuron, harbors *Enterococcus faecalis* (Firmicutes) [80]. According to previous research, several gut symbionts (*Aeromonas hydrophila* Chester (Aeromonadales: Aeromonadaceae), *Arsenophonus* nasoniae Pérez-Brocal et al. (Enterobacterales: Morganellaceae)*, Actinobacteria* sp., *B. cepacia* Yabuuchi et al., *Clostridium botulinum* Ermengem et al. (Eubacteriales: Clostridiaceae), *C. freundii*, *Enterococcus faecalis* Andrewes and Horde (Lactobacillales: Enterococcaceae), *Lachnospiraceae*, and *E. acetylicum* Collins) of insect species belonging to the orders Diptera, Coleoptera, Lepidoptera, and Hemiptera degrade different classes of insecticides such as Neonicotinoid, Carbamate, Organochloride, Methoprene, Benzoylurea, and Organophosphate [76].

The capacity of microbes to use insecticides as a carbon source is contingent on the coding of the metabolic systems required to deal with these substrates (Figure 4). The metabolism of pesticides is regulated by pH, nutritional availability, temperature, chemical concentration, and bacterial population number [87]. When xenobiotics spread quickly, the gut microbiota uses a variety of metabolic pathways to break them down [88]. For instance, the metabolic breakdown of imidacloprid is mediated by *P. aeruginosa* Migula [89] *Arsenophonus* sp. [90], *Stenotrophomonas maltophilia* Palleroni and Bradbury (Xanthomonadales: Xanthomonadaceae), *Ensifer meliloti* Casida (Hyphomicrobiales: Rhizobiaceae), and *Variovorax paradoxus* Willems et al. (Burkholderiales: Comamonadaceae) [91]. In another example, the insecticide thiamethoxam is digested by *Ensifer adhaerens* de-Lajudie et al. and *P. aeruginosa* Migula [92]. The major metabolic process consists in the conversion of its N-nitroimino group (=N-NO_2_) to N-nitrosimine/nitrosoguanidine (=N-NO, THX-II) and urea (=O; THX-III) metabolites [92].

## 9. Role of Gut Bacteria in Acquisition of Tolerance and Resistance

Insects have a multilayered defensive mechanism in their digestive tracts. This defensive mechanism aids in the host’s capacity to tolerate and fight microorganisms in the gut [93]. Resistance is the capacity to reduce the bacterial load such that it cannot affect the host’s health, while tolerance is the capacity to lessen the harmful effects of a given bacterial load on the host’s health [93]. Insect–microbial interactions in the gut are commonly mutualistic or communalistic and the host must limit any unfavorable effects of the local microbiota. Insects with larger bacterial populations are likely to have poor resistance but high tolerance to the bacteria in their guts [94]. Many insect midguts produce a protein–carbohydrate embedded in a peritrophic matrix that comprises chitin microfibrils [94]. The peritrophic matrix is semipermeable, allowing digestive enzymes, nutrients, and defense-related chemicals to flow through, while protecting the epithelial cell layer from direct exposure to pathogens or poisons [95]. This cuticle layer serves as a protective barrier for the epithelial cell layer of the foregut and hindgut [95]. This barrier between the epithelium and the lumen reduces bacterial effects on the host rather than lowering the bacterial burden in the gut [95].

According to Moreno-Garca et al. [96], a specific area of an insect’s stomach may have a low or high pH or include enzymes that attack peptidoglycan or lysozymes (PGN) hydrolases, which are parts of bacterial cell walls. Through modulating immunological responses, competing for locations, or producing inhibitory chemicals, the gut microbiota regulates vector capacity [97]. Insecticides and plant defenses could interact and boost insect immunity [98]. Plant secondary metabolites defend plants against herbivorous arthropods and play a vital role in insect resistance development [99]. Microorganisms in the stomach could digest substances, which may help with the removal or inactivation of hazardous substances found in insect diets. The gut microbiota of several social insects of the genera *Apis mellifera* L. (Hymenoptera: Apidae) and *Bombus terrestris* Latreille (Hymenoptera: Apidae) plays crucial roles in their health, ability to absorb nutrition, and defense against pathogens [100,101].

*Burkholderia cepacia,* a gut symbiont of *Riptortus pedestris* Fab. (Hemiptera: Alydidae), breaks down the pesticide fenitrothion and helps the organism survive in pesticide-contaminated soil [88]. The presence of gut microbiota has been demonstrated to boost the survival rate of the wasp *Nasonia vitripennis* Walker (Hymenoptera: Pteromalidae) under atrazine exposure [66]. It illustrates how the gut microbiota might enhance the insect host’s ability for adaptation, which has significant implications for the management of pests and pollinator insects. In this way, the gut microbes that are connected to insects benefit their host’s general health and wellness. The capacity of these microorganisms to give nutrition is their only fundamental function. Secondary bacterial symbionts aid in the detoxification of chemicals generated for herbivore defense [102], protect against heat stress, and improve the host’s immune response to entomophagy [103]. Moreover, the gut bacteria may detoxify xenobiotics by breaking down organic pesticide chemicals [104]. For instance, *Plutella xylostella* L. is immune to every kind of pesticide. The microbiota of insects includes a variety of taxa, including Bacilli, Flavobacteria, and Gammaproteobacteria, which aid in the enzymatic breakdown of the pesticide indoxacarb, which has xenobiotic-like properties [76].

## 10. Biological Significance of Gut Symbiotic Microfauna

For insect pests, the gut microbiome plays a crucial role. The gut microbiota is vital in food digestion, keeping harmful microorganisms out, producing necessary vitamins for the host, and performing metabolic detoxification [105]. In comparison with larger animals, insects have a relatively high capacity for adaptation. The bacteria in insect guts also help these creatures to have such tremendous adaptive capacities [105]. Little changes in the environment may cause the gut bacteria to interact with each other [105]. It has been shown via the discovery of quorum sensing that bacteria make pheromone-like molecules or autoinducers to interact with one another, which causes the synthesis of metabolic products that are dependent on population density. Quorum sensing is a typical occurrence in symbiotic bacteria and provides a comprehensive picture of the bacterial population and its interactions with the insect host [105]. The role of gut microbiota in pesticides/insecticides and phytotoxin degradation is summarized in Table 2 and Table 3.

### 10.1. Symbiont-Mediated Detoxification of Phytotoxin

Insects have been revealed to possess a variety of defense systems against both natural and synthetic poisons. These are mainly metabolic detoxification, mutations at target sites, behavioral avoidance, etc. All the genes responsible for these kinds of resistance have been encoded by the insect’s genome [113]. Repeated use of pesticides or repeated encounters with plant chemicals enhance the resistance power of insects through the mutation or rearrangement of their genomes. These resistant genes are transferred from one generation to the next and their future pest population becomes more resistant to the toxin [113]. Symbiont-mediated toxin degradation is important, because, when a trait incorporates rapidly into one insect generation via symbiont acquisition, this resistant trait is transmitted horizontally between insect species [113].

Insects contain various gene families, such as cP450 and GST, that are responsible for developing features for detoxification [115]. For example, *Myzus persicae* Sulzer (Hemiptera: Plataspidae) harbors the bacterium *Achromobactor xylosoxidans* Yabuuchi and Yano (Burkholderiales: Alcaligenaceae), which carries the gene carboxylesterase for the enzyme N-methylcarbamate hydrolase, that is involved in phytotoxin breakdown [114]. Gene duplication is the fundamental mechanism through which detoxication properties have developed [117]. Through horizontal gene transfer, microorganisms may also acquire or trade metabolic genes. Symbiotic bacteria serve as a genetic source of genome evolution in the insect host, since the detoxifying genes of associated bacteria may become incorporated into the host genome. The gut symbionts in phytophagous insects have various significant functions [114]. They contribute to the lignocellulose breakdown in xylophagous insects and create antimicrobial chemicals that defend the insect from pathogens. Certain insect gut microorganisms participate in plastic breakdown. Hence, the gut symbionts demonstrate a variety of detoxifying capacities [114,117].

For example, *Megacopta cribraria* Zhang and Wheeler (Hemiptera: Plataspidae) (Plataspids) is a notable pest of peas and soybeans. They may vertically transfer their obligatory *Candidatus Ishikawaella capsulata* Mpkobe (Enterobacterales: Enterobacteriaceae) symbiotic microorganisms by laying brown capsules carrying the microbes with their eggs, which are consumed by newly hatched nymphs [118]. These obligatory symbionts have an ode gene on one of their plasmids [118,119]. This ODE gene encodes for an enzyme called oxalate decarboxylase, which may degrade oxalate, a secondary metabolite found in plants that provides defense against herbivory [120]. Consequently, by detoxifying the plant’s secondary metabolites, this symbiont defends its host insect.

In another instance, the Brassicaceae family of plants includes several well-known vegetable crops such as cabbage and cauliflower, where the cabbage root fly (*Delia radicum* Linnaeus (Diptera: Anthomyiidae) feeds [110]. The detoxifying bacterial symbiont *A. baumannii* Brisou, which participates in the breakdown of the pesticide isothiocyanate, is found in the cabbage root fly [110]. Myrosine, an enzyme generated by cruciferous plants, facilitates the degradation of glucosinolates to create the poisonous isothiocyanates (ITC). In reaction to insect injury, the ITC activates a defense mechanism. The gut bacteria *A. baumannii* and *Pectobacterium* carotovorum Waldee found in cabbage root fly larvae are capable of detoxifying ITC [110]. Four strains of Gammaproteobacteria (*Pectobacterium* sp., *Serratia* sp., *Providencia* sp., and *A. baumannii* Brisou and Prévot) isolated from the guts of the root fly are capable of degrading ITC into less toxic chemicals. One plasmid, Drgb3, is present in the corresponding bacterial species. The Dag3 gene family includes the saxA gene [121], which encodes a new aromatic ITC hydrolase [110]. Thus, these gut bacteria reduce the amount of harmful isothiocyanate in the host’s gut, enhancing the host’s fitness. *Hypothenemus hampei* Ferrari (Coleoptera: Curculionidae) (also known as the coffee borer) is another pest of coffee and the main problem for growers of coffee beans all over the globe. A key stimulant found in coffee beans, caffeine, also serves as a defensive alkaloid allelochemical against herbivory. By employing its gut bacteria to degrade caffeine, insect *Hypothenemus hampei* (Coleoptera: Curculionidae) avoids the damaging effects of this compound [109]. The coffee borer’s intact gut microorganisms can completely deplete the caffeine in their diet, but antibiotic treatment destroys the gut flora’s capacity to decompose caffeine. The gut bacterium *P. fulva* of the coffee borer carries the ndmA gene, which encodes an enzyme called methylxanthine N1-demethylase that catalyzes the initial step in caffeine breakdown [122].

### 10.2. Symbiont-Mediated Detoxification of Insecticides

Numerous chemical pesticides have been industrialized to control agricultural pests. Insects can overcome synthetic poisons via a variety of defense mechanisms, including reduced penetration through a thicker cuticle, avoidance behavior, target-site mutation, and detoxication [123]. The insect genome is considered to encode all these resistance mechanisms [123]. Recent research demonstrated that symbiotic bacteria may mitigate the damaging effects of chemical pesticides on agricultural pests. *Bacillus cereus* Frankland, isolated from the digestive system of the moth *P. xylostella*, has shown strong breakdown and assimilation activities of the pesticide indoxacarb for use in the metabolism and development of insects [11]. According to van den Bosch and Welte [11], several insect gut microorganisms, including *Pantoea agglomerans* Gavini et al. (Enterobacterales: Erwiniaceae), *B. cereus*, and *Enterobacter asburiae* Farmer et al., breakdown acephate, an organophosphorus substance that inhibits acetylcholine esterase.

The oriental fruit fly, *Bactrocera dorsalis* Hendel (Diptera: Tephritidae), is a notorious pest of citrus fruits and horticulture crops. This fruit fly has shown resistance to the insecticide trichlorphon, an organophosphorus [124]. As a result, it has become a significant pest for food crops. A Gammaproteobacterium called *C. freundii*. has been found in the stomach of the resistant strain *B. dorsalis* (Diptera: Tephritidae); it hydrolyzes the poisonous chemical trichlorphon to produce the less harmful compounds dimethyl phosphite and chloral hydrate [71]. Hence, *Citrobacter* sp. is responsible for trichlorphon resistance in the fruit fly. The dangerous leguminous crop pest known as *Riptortus pedestris* Fab. (Hemiptera: Alydidae), which is widespread in East Asia, attacks soybeans. In the posterior area of the midgut of *Riptortus pedestris* Fab. (Hemiptera: Alydidae), there are many sac-like “crypts” that are mostly occupied by a symbiotic bacterium, *B. cepacia*. It has been observed that soil bacteria such as *Cupriavidus metallidurans* Mergeay et al. (Burkholderiales: Burkholderiaceae), *P. aeruginosa* Migula et al., *Sphingomonas paucimobilis* Yabuuchi et al. (Sphingomonadales: Sphingomonadaceae), *Corynebacterium diphtheriae* Lehmann and Neumann (Mycobacteriales: Corynebacteriaceae), *Arthrobacter globiformis* Conn and Dimmick (Micrococcales: Micrococcaceae), and *B. cepacia* decompose the organophosphorus pesticide fenitrothion (MEP) by removing its methanol derivative [125]. These bacteria serve a crucial function in the consumption and elimination of methanol, a toxic byproduct of MEP-degradation, to promote optimal growth and development [125]. *Burkholderia cepacia* is a predominant microorganism in soil that degrades MEP. As a result, this gut bacteria increases *R. pedestris’* resistance to fenitrothion [125].

### 10.3. Molecular Mechanism of Enzyme-Mediated Insecticide Detoxification

Insect-produced enzymes, such as cytochrome P450, esterases, and glutathione S-transferases, are responsible for insect defense against allelochemicals and other hazardous substances [79]. Detoxifying enzymes naturally exist in the insect body as a result of numerous physiological processes, acting on the target sites to neutralize various toxins that are present throughout the insect body [26,126]. Insecticide sensitivity at the specific target site and its detoxification via metabolic enzymes, including carboxylesterase, cytochrome P450, acetylcholinesterase, and glutathione S-transferase, are related to insect resistance to insecticides [127,128]. All these enzymes are essential for the detoxification of xenobiotics [127]; insects may use them as biological markers for insecticide detoxification [129]. Detoxifying enzymes (cytochrome P450 genes) have been found in *Solenopsis invicta* Buren (Hymenoptera: Formicidae) (red imported fire ants) during the detoxification of fipronil; an increase in resistance of up to 36.4-fold is seen when the ants are exposed to the drug [130]. These enzymes are engaged in several activities, such as metabolism and biosynthesis of invading species. Imidacloprid metabolism has been studied and evaluated using CYP6CM1 in *Bemisia tabaci* Gennadius (Hemiptera: Aleyrodidae) and P450 CYP6ER1 in *Nilaparvata lugens* Stal. (Hemiptera: Delphacidae) [131,132]. Insects use a variety of methods to remove xenobiotics from the stomach lumen. They can generate an acidic environment and provide an enzyme complex (monooxygenases and esterases) that may break down or modify the xenobiotic in preparation for elimination [108]. It has been proven that, in the gut lumen, microbial enzyme activity contributes to the degradation of pesticides eaten by the host. The hydrolysis of these chemicals enables the microbiota to grow by providing them with nutrients [133]. Hence, microbial enzymes may play an important role in the metabolization of insecticides of affected insects [87,108].

### 10.4. Gut Microbe-Mediated Nutrient Metabolism

The endosymbionts found in insect guts are crucial to the metabolism of nutrients. Nutritional contributions may come in a variety of ways, including the supply of vitamins, the acquisition of digestive enzymes, better digestion efficiency, and the improvement of the capacity to survive on suboptimal diets [79].

#### 10.4.1. Protein Metabolism

Several microbial species in the gut microbiota, including *Bacteroides fragilis* Veillon and Zuber (Bacteroidales: Bacteroidaceae), *C. botulinum*, and *L. acidophilus* Hansen and Mocquot, etc., include different proteases that are involved in the degradation of proteins. Peptide transporters, peptidases, proteinase activity, and the lactic acid bacteria (LAB) proteolytic system all operate collectively [134]. Protein hydrolysis by LAB is initiated by a cell envelope proteinase (CEP) that degrades the protein into oligopeptides, which are taken in by the cells through peptide transporters and then further degraded by several intracellular peptidases into shorter peptides and amino acids [134,135].

#### 10.4.2. Sugar Fermentation

Sugar is the most important primary source of energy. The significant metabolic step that is aided by gut microorganisms is the fermentation of sugar. It has been shown that the LAB’s proteolytic system has acquired the capacity to identify sugars such as cellobiose, fructose, glucose, and xylose [135]. Hence, it plays an essential function in lactic fermentation [134]. Symbiotic gut microorganisms may provide their host insects with crucial amino acids via sugar fermentation. It has been revealed that the gut symbiotic bacteria *B. aphidicola* can provide tryptophan and other important amino acids to their aphid hosts due to the shortage of key amino acids in their diet [134].

#### 10.4.3. Nitrogen Fixation

Symbiotic microorganisms are essential for the fixation of nitrogen. It is a crucial metabolic pathway for the growth and feeding of insects. According to reports, termites mostly obtain their nitrogen from intestinal microorganisms rather than food [136]. Termites’ feeding habits have an impact on how much nitrogen they can fix. Termites that mostly consume wood have a greater capacity for nitrogen fixation than termites that solely consume dirt. Intestinal bacteria such as *C. freundii* and *Enterobacter agglomerans* Beijerinck are crucial for nitrogen fixation in wood-eating termites [136]. Spirochetes, another type of gut bacteria in termites, play an important role in providing the carbon, nitrogen, and energy needs of termite nutrition through acetogenesis and nitrogen fixation [4]. For instance, *Blattabacterium cuenoti* Koga and Moran (Flavobacteriales: Blattabacteriaceae), a Gram-negative bacterium found in the fat body of cockroaches, has been implicated with nitrogen absorption and uric acid degradation [137]. In the order Coleoptera, nitrogen fixation has frequently been seen in beetles that feed on bark and woody detritus. The bacterial genus is primarily responsible for nitrogen fixation in bark beetles, *Dendroctonus ponderosae* Hopkins (Coleoptera: Curculionidae) [138]. *Candidatus Dactylopiibacterium carminicum* has been identified in two species of the Dactylopius coccus costa (Hemiptera: Dactylopiidae) scale insect that feed on plant sap [139].

#### 10.4.4. Cellulose Digestion

Insects that consume wood have cellulose-digesting bacteria in their digestive tracts. As an example, the mulberry leaf-eating *Bombyx mori* L. (Lepidoptera: Bombycidae) relies heavily on the digestive enzymes generated by the gut bacteria to break down carbohydrates including pectin, xylan, cellulose, and starch [140]. Termites are also active in the breakdown of cellulose into hexose and pentose oligomers and a variety of biofuel derivatives are catalyzed by bacteria in their stomachs. Current research has focused on termites and their potential to convert wood into biofuels through their intestinal microbiome, which has a significant cellulose degrading ability [17].

#### 10.4.5. Lipid Metabolism

Microbes in insect guts also have a significant influence on lipid metabolism. Gut microorganisms generate triglyceride metabolites for the host insects, which are utilized as a carbon and energy storage source. A group of crucial lipid compounds, called polyhydroxyalkanoates (PHAs), are generated by gut microorganisms in yellow mealworms. PHAs may be transformed by intestinal microorganisms into carbon and energy storage in the host insects [141]. A novel amino glycolipid that may stimulate the synthesis of the quinone reductase in host cells has been isolated from the stomach of the queen carpenter ant, *Camponotus japonicus* Mayr (Hymenoptera: Formicidae) [142].

#### 10.4.6. Vitamin Production

Microbes in the gut can deliver vital vitamins to their host, thus contributing to the maintenance of physical health. Gastrointestinal microbiotas convert nitrogen to ammonia, which is absorbed by gut microorganisms to provide vitamins for insect growth [48,143]. Vitamin B is synthesized by the intestines’ bacteria due to the water-soluble nature of the vitamin. Gastrointestinal bacteria play a crucial role in the supplementation of B vitamins for insect hosts with a deficiency of vitamin B. Moreover, it has been observed that *Wigglesworthia glossinidia* Aksoy et al. (Enterobacterales: Erwiniaceae), a symbiont of *Glossina brevipalpis* Newstead (Diptera: Glossinidae), needs several vitamins as cofactors for its own metabolism, including pantothenate, biotin, thiamine, riboflavin, FAD, pyridoxine, nicotinamide, and folate [23].

### 10.5. Insect Gut Bacteria-Mediated Plastic Degradation

In addition to the biological significance of the insect stomach in the digestion and absorption of nutrients, symbiotic bacteria also play other biological roles that are receiving attention. Due to their capacity to break down plastics, insect gut bacteria are receiving good interest in the field of bioremediation. The inappropriate usage and disposal of plastic waste results in several environmental pollutions [144]. If action is not taken, it is projected that the quantity of plastic waste will double over the course of the next 10 years, potentially having direct and indirect detrimental consequences on people [144]. Microorganisms capable of digesting plastic have been promoted as a solution to severe plastic pollution. In experimental settings, Indian meal moth *Plodia interpunctella* Hubner (Lepidoptera: Pyralidae) larvae commonly consume and chew polyethylene (PE) films, causing them to break down [145]. Yang et al. identified two PE-degrading gut bacteria from moth larvae, *B. subtilis* Ehrenberg (Bacillales: Bacillaceae) YP1 and *Enterobacter asburiae* Holmes et al., YT1 [145]. In particular, these two bacterial strains convert polyethylene (PE long chain) skeleton C-C groups into the -C=O- (carbonyl) group, which is regarded as a sign of PE breakdown [146]. Another instance is the bacterium *Ideonella sakaiensis* Yoshida et al. (Burkholderiales: Comamonadaceae), which can develop its growth on polyethylene terephthalate (PET) film and degrade film entirely in 6 weeks by secreting plastic-degrading enzymes termed PETase and MHETase [147,148]. Styrofoam, a polystyrene (PS) product, is consumed by the larvae of *Tenebrio molitor* L. (Coleoptera: Tenebrionidae) (yellow mealworm) and decomposed via depolymerization by their gut bacteria [149]. The *Tenebrio molitor*’s ability to degrade PS dramatically diminishes when antibiotics are used to eliminate the gut flora. In addition to free-living microorganisms, certain bacterial species found in insect guts are also capable of degrading plastics [146].

### 10.6. Gut Microbiota-Mediated Lignocellulose Digestion

An essential part of the plant cell wall, lignocellulose, is made up of a complicated network of cellulose, lignin, and hemicellulose. Animals can rarely digest lignocellulose on their own. Few insects have undergone independent lignocellulosic enzyme evolution [17]. For instance, higher termites and wood-eating cockroaches have certain stomach bacteria that make lignocellulose digestion easier [17,150]. In lower termites, cellulose and hemicellulose are jointly broken down by bacteria and flagellates via the use of enzyme cocktails in the termite hindgut [136]. Five *C. freundii* bacteria that break down cellulose have been isolated from the stomach of the root-eating white grub beetle, *Lepidiota mansueta* Harold (Coleoptera: Scarabaeidae), and show high levels of cellulolytic activity [21]. The bamboo snout beetle, *Cytrotrachelus buqueti* Guérin-Méneville (Coleoptera: Curculionidae) has some of the gut microbes such as *Lactococcus lactis* Tanner (Lactobacillales: Streptococcaceae), *S.*, *marcescens* Bizio, *Dysgonomonas gadei* Morotomi et al. (Bacteroidales: Dysgonomonadaceae), and *E. faecalis*. The main degrading enzymes for lignocellulose are carbohydrate-active enzymes (CAZymes), which have been discovered in the gut microbes of the bamboo snout beetle [32]. These CAZymes are responsible for the disintegration of bamboo cell walls, which aids in the development of the host insect. *Turicibacter faecalis* Andrewes and Horder (Erysipelotrichales: Turicibacteraceae), *Clostridium difficile* Hall and O’Toole, and *Novosphingobium* panipatense Gupta et al. (Sphingomonadales: Erythrobacteraceae) are present in the midgut of the wood-eating patent leather beetle *Odontotaenius disjunctus* Illiger (Coleoptera: Scarabaeidae), which degrades cellulose and xylan using a variety of enzymes, including glycoside hydrolases, lignin by polyphenol laccases, and Fe-Mn superoxide dismutase [32].

## 11. Potential of Gut Microbes in Pest Management

In addition to contributing to relationships with hosts, insect gut microbes also provide a novel resource for biotechnological applications. Mutualistic symbiosis is a significant field in which to look for bioactive substances and novel enzymes for potential uses in medicine, industry, and the environment [151]. The management of agricultural pests and disease vectors might be made easier by the regulation of parasitic symbiosis [151]. The development of insect–pest control techniques is possible with a proper understanding of the molecular mechanisms underlying insect–microbial interactions and their impact on hosts. There are several different methods available for employing symbionts to regulate insect vectors and pest control measures [4,151]. The loss of an insect’s essential symbionts may have a significant effect on the insect host. The ability of insects to perform several vital functions, such as metabolic needs, resistance to natural adversaries, and vectoring capacity, may also be hampered by manipulation of the gut bacteria. The genetically engineered gut bacteria are a potential component of future techniques for pest management. Before effective deployment, however, it is essential to achieve a thorough understanding of the persistence, colonization, and mechanism of transmission of the bacteria [4]. In the next section, we discuss numerous instances of insect–pest management approaches, some of which are now in use and others that will be developed in the future [1]. Typically, when employing biochemicals, there is a straightforward method to eradicate or disrupt insect symbiosis [35]. It has been shown that the use of antibiotics such as penicillin and tetracycline make tsetse flies infertile by harming the obligatory mutualist *Wigglesworthia glossinidia* Dale and decreases adult tick reproduction by reducing their symbiont load [152]. In vitro, the use of antibiotics may completely eradicate endosymbionts, which would shorten an insect’s lifespan and lower the number of pests. Therefore, using antibiotics for pest control in the field is not a viable alternative [23].

Antimicrobial peptides (AMPs) have also been used in certain instances to control insect symbionts. The positively charged surface of the AMPs may attach to the negatively charged surface of the microbial surface through charge–charge interactions, interfering with the integrity of the bacterial cell wall [153]. According to Fieck et al. [154], cecropin is an antimicrobial peptide that targets protozoan parasites such as *Plasmodium vivax* Grassi and Feletti (Haemosporida: Plasmodiidae) and *Trypanosoma cruzi* Chagas. Transgenic expression of cecropin in *Anopheles gambiae* Meigen (Diptera: Culicidae) has been found to reduce *Plasmodium berghei* Vincke and Lips (Haemosporida: Plasmodiidae) oocysts by 60% [155]. It has been discovered that the transgenic co-expression of cecropin-A and defensin-A in *Aedes aegypti* L. (Diptera: Culicidae) effectively prevents the spread of the *Plasmodium* parasite [142]. In addition, it has been shown that the experimental substitution of the main symbiont *B. aphidicola* with a different genotype by microinjection alters the amount of temperature tolerance of the pea aphid pest [156].

Incompatible insect technique (IIT) is another strategy employed for pest control. The process for IIT often includes cleaning the surface of the insect eggs to remove microbes that are deposited maternally. This method has further been developed by administering antibiotics to adult or larval insects together with their diet or other microbially rich items [157]. Arthropods often harbor the *Wolbachia pipientis* Hertig and Wolbach endosymbiont, which is vertically transferred. It may infect more than 60% of all insects [42]. Through controlling cytoplasmic incompatibility (CI) and male killing activities, this endosymbiont may influence host fertilization, parthenogenesis, and host reproduction [41]. To eliminate mosquitoes and other insect pests, the incompatible insect approach uses Wolbachia-induced CI [41]. Male Wolbachia-infected insect populations are commonly released in IIT to compete with natural insect populations [158]. Effective paratransgenic strategies for controlling insect pests mostly rely on the genetic design, the selection of microorganisms, and the use of the treated insects. In addition, the IIT method enables the use of parasitic symbiosis with entomopathogens in significant approaches to the management of disease and pest-carrying insects [158].

## 12. Role of Gut Symbiont in Insecticide Resistance and Possible Management Strategies

One of the biggest issues in agriculture is insecticide resistance. The microbiota in insect guts may reduce the toxicity of insecticide and produce a variety of detoxifying enzymes, including cytochrome P450, carboxylesterase, GST, and acetylcholinesterase [127]. The modulation of enzymes and the expression of several microbial detoxification genes in insect guts results in the development of resistance against pesticides [159]. In a study, Kikuchi et al. [85] discovered that, following the application of insecticides, insecticide-degrading bacteria are concentrated in agroecosystems. As a result, when insect pests appear in agroecosystems, they rapidly acquire the symbiont and develop resistance. Other studies have shown that *Riptortus pedestris* Fabricius (Hemiptera: Alydidae) are also connected to gut symbionts known as *B. cepacian*, which may degrade fenitrothion and increase the pests’ degree of pesticide resistance [160]. Early research has demonstrated that repeated exposure to fenitrothion causes a rapid increase in fenitrothion-degrading *Flavobacterium* sp., *B. cepacia*, and *P. aeruginosa* Migula et al., in agricultural field soils. These bacteria can degrade fenitrothion into 3-methyl-4-nitrophenol with very little insecticidal activity and metabolize the degrading product as a good source of carbon for their growth [161]. Hence, the gut symbiont is crucial for an insect’s ability to build resistance.

It has been shown that the use of antibiotics is particularly successful in eradicating or controlling the targeted gut bacteria, making it an effective technique for insect and pest management. For example, research into the roles of the melon fruit fly, *Bactrocera cucurbitae* Coquillett (Diptera: Tephritidae), shows that symbiotic bacteria might provide a new target for a unique sterilization-based control approach [162]. The melon fruit fly (*Bactrocera cucurbitae*) larvae are treated with oxytetracycline and sulfanilamide, which kill the symbiotic bacteria in the mycetocytes of the midgut area and eventually lower larval survival rates [162]. When copper carbonate is applied topically to *Bactrocera cucurbitae* Coquillett (Diptera: Tephritidae), the gut symbiotic bacteria suffer significant mortality [163]. Another example is the cytoplasmic incompatibility caused by the species *Wolbachia pipientis* Hertig and Wolbach, which results in male killing or feminization in its hosts. Hence, these bacteria are helpful for controlling insect pests. *Anopheles stephensi* Liston (Diptera: Culicidae), a significant malaria vector, has been managed with *Wolbachia* pipientis Hertig and Wolbach (Family: Anaplasmataceae) [164]. For instance, *C. freundii* Werkman and Gillen, a gut symbiont of the fruit fly *Bactrocera dorsalis* Hendel (Diptera: Tephritidae), is essential for the breakdown of trichlorfon and aids in the development of pesticide resistance [164]. The fluorescence in situ hybridization (FISH) and polymerase chain reaction (PCR) detection techniques indicate that *C. freundii* is accumulated in rectal pads associated with the female ovipositor and that the symbiont is transferred vertically through egg surface contamination [165]. This opens up new research directions for the management of *Bactrocera dorsalis* as a result of the discovery and antibiotic treatment of this gut symbiont [24]. The symbiotic bacterium *P. putida* Migula, engineered with toxin genes from the fruit fly *Bactrocera* tau Walker (Diptera: Tephritidae), may be very beneficial for pest control [166].

## 13. Biotechnological Applications Based on Insect–Microbe Interactions

Several biotechnological applications take advantage of the interactions between insects and microbes. The biotechnological application based on insect symbionts includes four distinct features including environmental applications, industrial applications, therapeutic applications, disease and pest control applications, and prospective environmental applications.

### 13.1. Industrial Application

Termite gut symbiotic bacteria may synthesize several enzymes to break down lignocellulosic plant biomass components, including mannanase, glucanase, and cellulase, and provide valuable resources for a variety of industrial applications [150]. Termite gut symbionts generate β-D and endo-1,4-D-glucanases, enzymes involved in the saccharification of cellulose in cellulosic biofuels [167]. EMB156, the gut bacteria of *Bombyx mori* Linnaeus (Lepidoptera: Bombycidae) Bombyx, produces alkaline-tolerant alpha amylase and xylose isomerase enzymes that are very efficient for lactate fermentation, producing a compound for potential biotechnological use [168].

### 13.2. Clinical Applications

Many therapeutic applications of insect–microbe interactions are carried out. Symbiotic relationships between insects and bacteria play a significant role in the creation of novel chemicals and enzymes with medicinal and commercial promise. Symbiotic relationships between insects and microbes result in a variety of protective chemicals that may be employed to fend off parasites, predators, and diseases in the microbiome [169]. For instance, several kinds of termite gut have yielded isolates of the bacteria *Streptomyces coelicolor* Hopwood (Streptomycetales: Streptomycetaceae)*, Mycobacterium tuberculosis* Lehmann and Neumann (Mycobacteriales: Mycobacteriaceae), and *Kitasatospora setae* Omura et al. (Kitasatosporales: Streptomycetaceae) [169]. *Acromyrmex octospinosus* Reich (Hymenoptera: Formicidae), a leaf-cutting ant, produces the antifungal candicidin, which is isolated from the exoskeletons of *S. coelicolor* [170]. This antifungal chemical is effective against *Candida albicans* Siegel, an important human pathogen. The rove beetle, *Paederus fuscipes* Fabricius (Coleoptera: Staphylinidae), has a Gammaproteobacterial symbiont that generates the polyketide toxin pederin, which inhibits protein production and functions as a powerful antitumor agent [171]. In mosquito control issues, *Wolbachia*’s capacity to prevent pathogens has been proven to be a significant characteristic [172]. Medically significant arboviruses are transmitted by the *Aedes aegypti* Linnaeus (Diptera: Culicidae) mosquito. It does not naturally contain *Wolbachia*, but, when transinfected with *Wolbachia*, it exhibits significantly reduced resistance to the Zika virus, dengue, chikungunya, and yellow fever [173,174]. Another similar approach, called paratransgenesis, which uses genetically altered microorganisms to produce desired effects in insects, has been employed recently. The genetically engineered gut bacterium *Rhodococcus rhodnii* of the triatomine bug (*Rhodnius prolixus* Stal et al. (Hemiptera: Reduviidae) acquires the capacity to produce effector molecules (cecropin A and a pore-forming molecule) against the Chagas disease-causing protozoan parasite *Trypanosoma cruzi* Chagas (family Trypanosomatidae). Antitrypanosome single-chain antibody transformation of the symbiont may demonstrate a significant decrease in parasite burden [175]. Utilizing bacteria and fungi obtained from the midguts and ovaries of mosquitoes, paratransgenic techniques have also been used to stop the spread of the *Plasmodium vivax* Grassi and Feletti (Plasmodiidae), parasites that cause malaria.

The Gram-negative bacterium *Asaia bogorensis* Favia et al. (Rhodospirillales: Acetobacteraceae), which lives in mosquito midguts, has been chosen for paratransgenesis against *Plasmodium berghei* Vincke et al. [176]. The siderophore receptor gene and antiplasmodial effector genes are combined to create the Asaia strains. These genetically engineered genes, which include the scorpine antimicrobial peptide and a synthetic anti-Pbs21 scFv-Shiva1 immunotoxin, are effective against the *Plasmodium berghei* Vincke et al. ookinete surface protein 21- Shiva1 fusion protein. After feeding on the modified *Asaia* and being exposed to *Plasmodium berghei*-infected blood, the development of the parasite in *Anopheles stephensi* Liston (Diptera: Culicidae) mosquitoes is dramatically suppressed [177]. Insect defense symbionts create a sequence of antimicrobial ribosomal peptides that, soon, may replace current antibiotics.

### 13.3. Environmental Applications

Symbiotic bacteria have a variety of possible uses, making them interesting sources for bioremediation. For many years, plastic items, including polyethylene (PE), have been regarded as nonbiodegradable. Nevertheless, *E. cloacae* Fab. (Enterobacterales: Enterobacteriaceae) and *B. subtilis* found in the intestines of the Indian meal moth *Plodia interpunctella* Hubner (Lepidoptera: Pyralidae) may degrade 6.1% to 10.7% of the PE film [146]. The PE polymer has also been observed to be degraded by the larvae of the wax moth, *Galleria mellonella* L. (Lepidoptera: Pyralidae). There is significant potential for bioremediation using insects and mutualistic symbiotic microorganisms.

## 14. Conclusions and Future Perspective

Insects often possess symbiotic microbiota in their guts, which provide evolutionary advantages, including indigestible diet digestion, protection from antagonists, and detoxification of various toxins. Natural selection throughout evolutionary time has bred these special abilities into gut symbionts, which are naturally sophisticated. The invisible but pivotal insect partners offer great potential for the study of ecological evolutionary biology and industrial applications, as we are only just beginning to understand the gut microbiota of this enormously diverse animal group.

Increasingly: it is understood that the gut endosymbionts of insects are important sources for future pest control programs, with tremendous biotechnological potential. The variety of interactions between bacteria and insects reveals the processes for regulating and controlling insect populations from an agricultural perspective. By employing transgenic versions of the related microorganisms or by introducing pathogenic organisms that would compete with, replace, or regulate the symbionts, it may be possible to manipulate the symbiotic bacteria. These more recent nonchemical strategies could be useful for managing an insect pest colony that is rapidly increasing. In recent years, there has been a surge of interest in insect–symbiont interactions and symbiont-mediated toxin breakdown. Contemporary research on gut symbionts will provide fresh insights into how to effectively manage insect pests via symbiont management, including how to interrupt the close relationships and regulate the detoxifying symbionts. Yet, in most connections, the method of gut–symbiont transfer is still unknown.

## Figures and Tables

**Figure 1 microorganisms-11-02665-f001:**
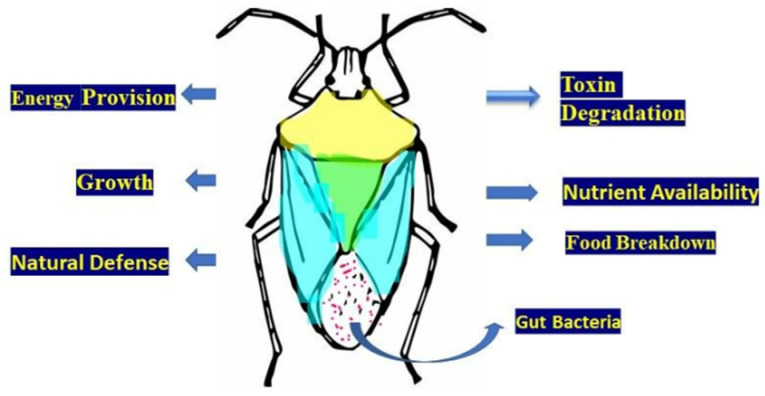
Functional impact of gut microbiota on insect physiology.

**Figure 2 microorganisms-11-02665-f002:**
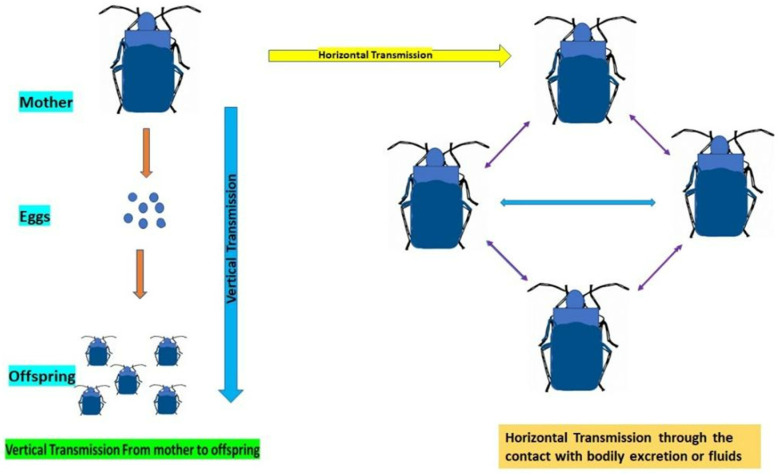
Mechanism of transmission of gut symbionts.

**Figure 3 microorganisms-11-02665-f003:**
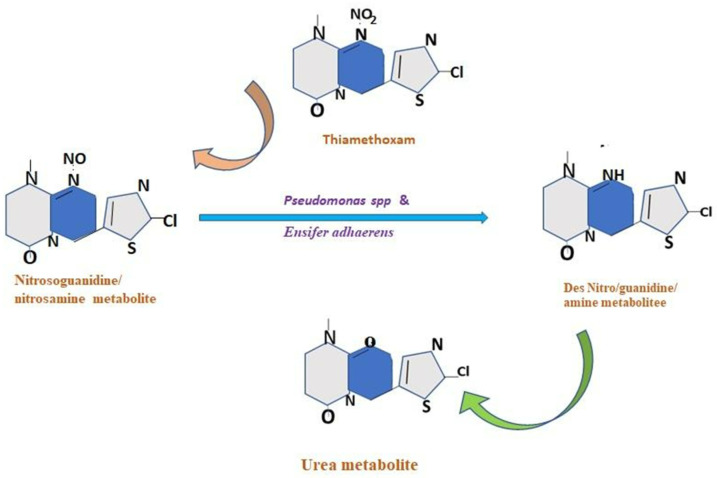
*Ensifer adhaerens* and *Pseudomonas* spp. metabolized the thiamethoxam pesticide. Metabolic routes for bacterial degradation of the insecticide thiamethoxam.

**Figure 4 microorganisms-11-02665-f004:**
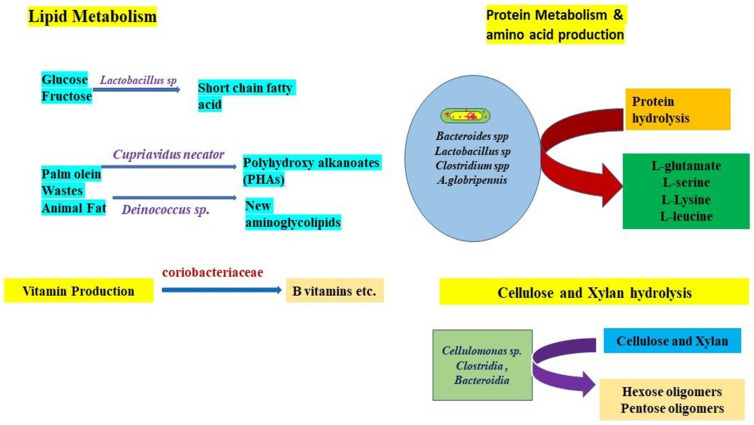
Gut bacteria-mediated nutrient metabolic process in insects.

**Table 1 microorganisms-11-02665-t001:** List of gut microbiotas reported from insects.

Insect Order, Common Name, Species Name	Bacterial Species	Type of Interaction	Phenotype	References
Hemiptera				
Blood sucking bug: *Rhodnius prolixus* Stal et al. (*Reduviidae*)	*Rhodococcus rhodnii* Tsukamura et al.	Gut symbiont/Commensal	Amino acid synthesis	[23]
Sap-sucking insects: Aphids, *Acyrthosiphon pisum* Harris, *Schiaphis graminum* Rondani et al.	*Buchnera aphidicola* Munson et al.	P-endosymbiont	Confers host defense against natural enemies, parasitic wasps	[33]
Aphids, *Acyrthosiphon pisum* Harris (*Aphididae*)	*Hamiltonella defensa* Moran et al.	S-symbiont	Confers host defense against natural enemies, parasitic wasps	[33]
Sap-sucking insects: Psyllids, *Pachypsylla venusta* Thomas et al. (*Psyllidae*)	*Carsonella ruddii* Thao et al. (y-proteobacteria)	Endosymbiont	Essential nutrients, possibly amino acids	[34]
Sap-sucking insects: mealybugs, *Planococcus citri* Risso et al.	*Tremblaya princeps* (β-proteobacteria)	Endosymbiont	Probably amino acid	[35]
Orthoptera				
Grassland locusts, *Myrmeleotettix palpalis* Zubovski, 1900	Serratia marcescens strain HR-3 (y-proteobacteria)	Pathogen	Paralysis induced by insecticidal metalloprotease.	[36]
Anoplura				
Human body louse, *Pediculus humanus* L. (*Pediculidae*)	*Rickttsia prowazekii* da Rocha-Lima et al.	Obligate Intracellular		[37]
Coleoptera				
Rice weevil, *Sitophilus oryzae* L. (*Curculionidae*)	P-endosymbiont SOPE (y-proteobacteria)	P-endosymbiont	Vitamin synthesis and in fluence mitochondrial respiration in the host	[38]
Neuroptera				
Antlion *Myrmeleon bore* ( *Myrmeleontide*)	*Enterobacter aerogenes*, *Bacillus cereus*, *B. sphaericus*, *Morganella morganii*	Temporal association	Pathogens for other insect species prey of the antlion.	[39]
Siphonaptera				
Human North America Flea, *Oropsylla montana* (*Ceratophyllida*)	Yersinia pestis Lehmann and Neumann, (γ-proteobacteria)	Vector	Transmission of mammalian and human pathogen	[40]
Diptera				
Tsetse fly, *Glossina* spp. (*Glossinidae*)	*Wigglesworthia glossinidia* Dale et al. (γ-proteobacteria)	Symbiont	Cytoplasmic incompatibility	[24]
Fruit fly, *Drosophila melanogaster* Meigen (*Drosophilidae*)	*Sodalis glossinidius* Dale and Maudlin (γ-proteobacteria)	Symbiont	Cytoplasmic incompatibility	[41]
Tsetse fly, *Glossinia brevipalpis* Newstead (*Glossinidae*)	*Wolbachia pipientis* Hertig and Wolbach (α-proteobacteria)	P-endosymbiont	Essential for fly fertility	[42]
Lepidoptera				
Tobacco horn worm, *Manduca Sexta* L. (*Sphingidae*)	*Photorhabdus luminescens* Thomas et al.	Pathogen	Several toxins with oral and injectable toxicity	[43]
Wax moth, *Galleria mellonella* L. (*Pyralidae*)	(γ-proteobacteria) *Xenorhabdus nematophilus* Thomas and Poinar.	Pathogen	Xpt and Xax Toxins	[44]
Hymenoptera				
Carpenter ant, *Camponotus floridanus* Buckley (*Formicidae*)	*Blochmannia floridanus* Blochmann (γ-proteobacteria)	Nonessential endosymbiont	Improves viability of host pupae	[45]

**Table 2 microorganisms-11-02665-t002:** Insect gut microbiota involved in degradation of insecticides.

Insect Pests	Gut Microbiota	Insecticides	Reference
*Drosophila melanogaster* Meigen (*Drosophilidae*)	*Acetobacter* spp. Beijerinck et al., *Lactobacillus acidophilus*, *L. plantarum* Orla-Jensen	Neonicotinoid	[106]
*Rhagoletis pomonella* Walsh (*Tephritidae*) *Anopheles stephensi* Liston (*Culicidae*)	*Exiguobacterium* sp. Collins et al., *Aeromonas* spp. Stanier et al., *P. putida* Migula, *Citrobacter freundii* Werkman and Gillen	Organochloride, Organophosphates	[90]
*Rhagoletis pomonella* Walsh *Aedes* spp. and *Anopheles gambiae* Meigen (*Culicidae*)	*Lysinibacillus* spp., Meyer and Neide, *Staphylococcus* spp. Rosenbach et al., *P. melophthora*, *Clostridium botulinum* Van Eminem	Carbamate, Methoprene.	[107]
*Spodoptera frugiperda* Smith (*Noctuidae*)	*Microbacterium arborescens* Imai et al., *Staphylococcus sciuri*, *Enterococcus mundtii* Collins et al.	Benzoylurea	[108]

**Table 3 microorganisms-11-02665-t003:** Insect gut microbiota involved in phytotoxin and insecticide degradation and their mode of action.

Insect Species	Gut Bacteria Present	Action on Phytotoxin	Enzyme Involved in Degradation	Name of the Gene	Reference
*Drosophila melanogaster* Meigen (Diptera: Drosophilidae)	*Pseudomonas fulva* Iizuka and Komagata	Caffeine	Methylxanthine N1-demethylase	GST, P450	[109]
*Trichoplusia ni* Hübner (Lepidoptera: Noctuidae)	*Pectobacterium* sp. Jones et al.	Isothiocyanates	Metal-dependent beta-lactamase	GST	[110]
*Dendroctonus ponderosae* Hopkins (Coleoptera: Curculionidae)	*Pseudomonas* sp. Migula et al.	Terpenes	Diterpene acid degradation pathway	GST	[111]
*Drosophila melanogaster* Meigen	*Enterobacter asburiae* Farmer et al.	Phenols	Oxygenase, Isomerase, Transferase	GST	[112]
*Myzus persicae* Sulzer (Hemiptera: Aphididae)	*Escherichia coli* O157	Glycosides	6-Phospho-beta-glucosidase	GST	[113]
*Myzus persicae* Sulzer	*Achromobactor* sp. Yabuuchi and Yano	Carbamates	N-methylcarbamate hydrolase	Carboxylesterase	[114]
*Drosophila melanogaster* Meigen	*Burkholderia cepacia*	Organophosphates	Organophosphate hydrolase	GST	[109]
*Anopheles gambiae* Meigen (Diptera:Culicidae)	*Sphingobium japonicum* UT26. Pal et al.	Organochlorines	Lin pathway	GST, P450	[115]
*Cimex lectularius* Linnaeus (Hemiptera: Cimicidae)	*Variovorax boronicumulans* Miwa et al.	Neonicotinoids	Nitrile hydratase	Esterase, GST, P450	[116]
*Anopheles gambiae* Meigen, *Musca domestica* L. (Diptera: Muscidae)	*Sphingobium* sp. Pal et al.	Pyrethroids	Carboxylesterase	P450	[66]

Gene abbreviations: GST—glutathione S-transferases; P450—cytochrome P450; COE—carboxylesterase.

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
