# Peer review of "Insect Microbial Symbionts: Ecology, Interactions, and Biological Significance"

_microorganisms, 2023, doi:10.3390/microorganisms11112665_

Round 1
Reviewer 1 Report
This is an interesting review article on very important topic which described in detail the occurrence of microbiota in the gut of insects as their diversity significantly affects the insect’s physiology. The authors tried to throw light on the pesticide resistance management and how microbiota may cause resistance to insecticides. This is a comprehensive article and authors tried to address all relevant information including the eye-catching graphics and tabulated good information in the article.
Albeit, I can indicate some grammatically weak sentences but it is suggested the authors should seek help from English native colleague to read the ms for fluency in language.
Keeping in view the importance of topic and hard efforts of authors convinced me to accept this review article in its current form for publication in Microorganisms!
Author Response
Dear Reviewer
Thank you for the comments for the improvement of language. We revise the English of whole MS and removed any Grammatical and syntax errors, now we hope that language us suitable for publication.
Reviewer 2 Report
Dear Authors,
The text is interesting but there is lack of references. Also latin names should be used properly, in the entire text.
Please find my comments in the attached file.
Best regards

Author Response
Response 1: Thank you for the comments. We have added relevant references in the revised MS. Also we have corrected all the binomial names in the italic alongwith authors name. We also addressed all comments mentioned on the MS by reviewers. Now we hope that MS is suitable for publication.
Round 2
Reviewer 2 Report
Dear Authors,
Thank you for enhancing the manuscript with the references. The latin names need further changes. They should look like this the first time they are mentioned in the manuscript: e.g., Pseudomonas syringae Van Hall (Pseudomonadales: Pseudomonadaceae), and P. syringae all the other times except when the are the first words of a sentence (again like Pseudomonas syringae). Please check and change them throughout the manuscript.
Best regards
Author Response
Comment 1. Thank you for enhancing the manuscript with the references. The latin names need further changes. They should look like this the first time they are mentioned in the manuscript: e.g., Pseudomonas syringae Van Hall (Pseudomonadales: Pseudomonadaceae), and P. syringae all the other times except when the are the first words of a sentence (again like Pseudomonas syringae). Please check and change them throughout the manuscript.
Response. Thank you for your further comments to improve MS. We have revised the MS as suggested. Somewhere we put full genus name to avoid confusion with other genus starting with simmilar letter.